# A ConvNext-Based and Feature Enhancement Anchor-Free Siamese Network for Visual Tracking

Qiguo Xu [1], Honggui Deng [1],*, Zeyu Zhang [1], Yang Liu [1], Xusheng Ruan [1] and Gang Liu [1,2]

1. School of Physics and Electronics, Central South University, Changsha 410017, China; 202211045@csu.edu.cn (Q.X.); 192211038@csu.edu.cn (Z.Z.); 192211037@csu.edu.cn (Y.L.); 202212071@csu.edu.cn (X.R.); 162201003@csu.edu.cn (G.L.)
2. College of Information Science and Engineering, Changsha Normal University, Changsha 410199, China
* Correspondence: denghonggui@csu.edu.cn; Tel.: +86-199-7499-4794

**Abstract:** Existing anchor-based Siamese trackers rely on the anchor's design to predict the scale and aspect ratio of the target. However, these methods introduce many hyperparameters, leading to computational redundancy. In this paper, to achieve outstanding network efficiency, we propose a ConvNext-based anchor-free Siamese tracking network (CAFSN), which employs an anchor-free design to increase network flexibility and versatility. In CAFSN, to obtain an appropriate backbone network, the state-of-the-art ConvNext network is applied to the visual tracking field for the first time by improving the network stride and receptive field. Moreover, A central confidence branch based on Euclidean distance is offered to suppress low-quality prediction frames in the classification prediction network of CAFSN for robust visual tracking. In particular, we discuss that the Siamese network cannot establish a complete identification model for the tracking target and similar objects, which negatively impacts network performance. We build a Fusion network consisting of crop and 3Dmaxpooling to better distinguish the targets and similar objects' abilities. This module uses 3DMaxpooling to select the highest activation value to improve the difference between it and other similar objects. Crop unifies the dimensions of different features and reduces the amount of computation. Ablation experiments demonstrate that this module increased success rates by 1.7% and precision by 0.5%. We evaluate CAFSN on challenging benchmarks such as OTB100, UAV123, and GOT-10K, validating advanced performance in noise immunity and similar target identification with 58.44 FPS in real time.

**Keywords:** visual tracking; ConvNext network; features enhancement; anchor-free

## 1. Introduction

Visual tracking is a significant research problem in the field of computer vision. As long as the target state of the initial sequence frame is acquired, the tracker needs to predict the target state of each subsequent frame [1,2]. Visual tracking is still challenging in practical applications because the target is in various complex scenes such as occlusion, fast motion, illumination changes, scale changes, and background clutter [3,4].

The current popular visual tracking methods focus on the Siamese network [5]. These approaches typically consist of a Siamese backbone network for feature extraction, an interactive head, and a predictor for generating target localization. The Siamese network defines the visual tracking task as a target-matching problem and learns the similarity mapping between the template and search images through the interactive head. However, the tracker cannot predict the target scale effectively since a single similarity mapping usually contains limited spatial information. CFNet [6] proposed combining the filter technology with deep learning, and each frame would combine the previous template to calculate a new template. This approach ensures effective tracking when scale changes are large. Siamese FC et al. [7,8] propose matching multiple scales in the search region to determine the target

scale's variation. Nevertheless, these trackers require repeated computations and are time-consuming. SiamRPN [9] utilizes the Siamese network for feature extraction and region proposal networks(RPNs) [10,11] for classification and detection. The application of RPNs avoids the time-consuming step of generating multi-scale feature maps when predicting target scales. Later works such as DaSiam [12], CSiam [13], and SiamRPN++ [14] improved SiamRPN in terms of the dataset, network architecture, and network depth, which continuously promote tracking performance. However, RPN always requires many anchor boxes with redundant hyperparameters, i.e., the number, scale, and aspect ratio of candidate proposal boxes, which leads to computational and memory storage overload, making these methods inefficient in the training and testing phases. Secondly, Transformer has recently been widely applied in improving visual tracking algorithms. The Transformer has been introduced as a more robust interactive head for Siamese network-based trackers for providing information interaction. Researchers [15] propose using the Transformer to model the global temporal and spatial feature dependence between the target object and the search area. Other researchers [16] built Transformer architectures to explore their sequential context information. Under the guidance of the above, TansT [17] proposed the context-information-enhancement module based on self-attention and the feature enhancement module based on cross-attention. In addition SwinTrack [18] employs Transformer as the backbone network for feature extraction, allowing entire interactions between tracking target objects and search areas with a feature fusion strategy designed using Transformer. Tracking accuracy is advanced to a new level. However, these transformers are highly customized and well-designed, with complex frameworks that make it challenging to incorporate into a more general system or generalize various intelligent tasks. These networks are not lightweight enough for practical applications and require many computing resources for training.

Although the above algorithms based on anchor design and Transformer achieved specific achievements in performance, they are challenging to train and have high computational complexity. Therefore, the tracking real-time performance and the practical application are poor. We implement a ConvNext-based anchor-free Siamese Network (CAFSN) to achieve high-performance visual tracking in an end-to-end manner. CAFSN consists of three parts, namely a feature extraction network, a 3D C-Max fusion feature fusion network, and a classification and regression prediction network. Similar studies such as SiamBAN [19] use Resnet50 [20] as the feature extraction network. However, ConvNext [21] is the most advanced backbone for feature extraction networks. In order to improve network performance, the feature extraction network is based on ConvNext, which consists of a series of convolutions to obtain the local information by applying a specific size of convolution to a local image region. The convolution has translation invariance and can significantly respond to similar objects even at different locations [22]. In the field of target detection, some researchers [23,24] proposed that the context enhancement module (CEM) combine feature maps from multiple scales to leverage local and global context information. This inspired us to research and propose a 3D C-Max fusion feature fusion network. This module uses 3DMaxpooling to select the highest activation value to improve the difference between it and other similar objects and integrates features of different levels to promote the model's performance. Finally, we decompose the prediction into a classification problem and a regression task, where the regression task is to predict a relative bounding box corresponding to each position. In the classification task, points far from the target center tend to produce low-quality prediction boxes [25]. We design a central confidence branch based on the Euclidean distance to remove outliers and improve the network's overall performance.

As shown in Figure 1, our proposed CAFSN uses an end-to-end online training and offline tracking strategy, outperforming the advanced tracker SiamRPN in tracking. Ablation experiments verify the effectiveness of each module proposed in this paper. Our main contributions are as follows:

1. To reduce the computational and storage resources, we improve the tiny-ConvNext feature extraction network and introduce it for visual tracking. This paper proposes a CAFSN with a simple structure and powerful performance.

2. This paper proposes a feature fusion network combining cropping and 3D Max pooling (3D C-Max) with better abilities to discriminate similar targets.

3. A central confidence branch based on euclidean distance is proposed to suppress low-quality prediction frames, which improve the network's robustness and precision.

4. Our proposed tracker achieves advanced performance with an average speed of 58.44 FPS (Frames Per Second) on a GOT10k benchmark.

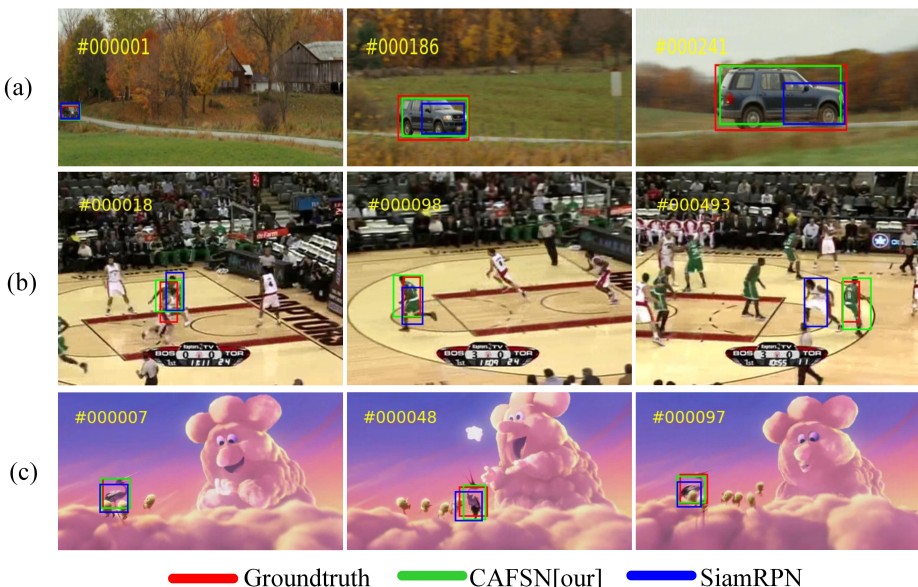

**Figure 1.** Results of our tracker and SiamRPN on OTB100. (**a**) is visualization of tracking result in CarScale sequences. (**b**) is visualization of tracking result in basketball sequences. (**c**) is visualization of tracking result in Bird2 sequences.

## 2. Related Works

This section briefly reviews the work of the backbone network, Siamese network, and detection model in the tracking field.

### 2.1. Backbone on Tracking

Convolutional neural networks (CNNs) have been widely implemented for target classification and detection tasks with excellent performance in recent years. Researchers have been encouraged to design CNN backbone networks using the Siamese network as a framework to achieve high-performance visual tracking. The most popular backbone networks among CNN trackers [8,9,26–28] in recent years are AlexNet [29], VGGNet [30], and ResNet [20]. Then, guided by the principles of CNN, the Transformer has been widely introduced into the vision field since 2020. Vision Transformer(ViT) [31,32] demonstrate a pure transformer applied directly to sequences of image patches can achieve excellent performance. Swin-Transformer [33] perfects on ViT by introducing patch merging, making the patch window more extensive and increasing the receptive field. The ConvNext network is based on some progressive ideas of the Transformer network to adjust to the existing classical ResNet network, which introduces some of the latest ideas and technologies of the Transformer network into the existing modules of the CNN network to improve the CNN network's performance. In image recognition and classification, the accuracy rate increases to 82.0%. Therefore, we improve its network stride and receptive field for introducing the most advanced CNN network in the proposed CAFSN. Figure 2 shows the specific details of the ConvNext network, which consists of one stem and four stages. The roles of the stem

and the downsample layer are to adjust the number of channels, and the role of ConvBlock is to deepen the network.

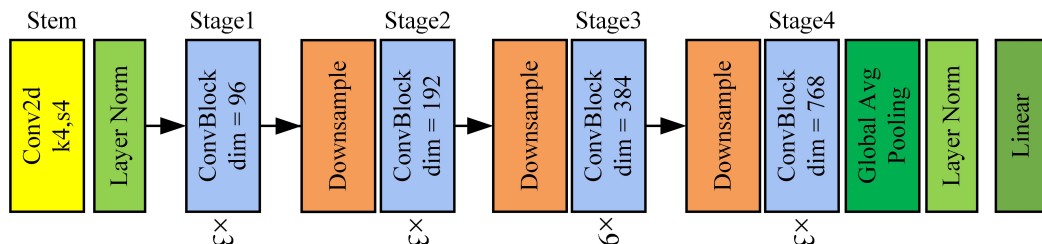

**Figure 2.** The specific details of the ConvNext network. Same color means same structure, Yellow indicates single convolution. Grass green indicates normalization. Orange indicates downsampling module. Blue indicates the ConvBlock module. Green indicates pooling module.

### 2.2. Background On Siamese Tracking

The fully convolutional Siamese tracker SiamFC is the basic framework discussed. The standard Siamese network input is a sample image $z$ and a search image $x$ pair. These two images are from the same video sequence, the interval difference between them cannot exceed $T$ frames, and the size of $z$ is minor than $x$. Generally, the size of $z$ is $127 \times 127$, and $x$ is $256 \times 256$. The Siamese network performs the same transformation on inputs $z$ and $x$ and then computes the similarity of all panning windows on a dense grid. The similarity function uses a mutual correlation with the following equation:

$$f(z, x) = \varphi(z) \times \varphi(x) + b1 \tag{1}$$

where $b1$ denotes the bias term at each position, and Equation (1) is equivalent to an exhaustive search for pattern $z$ on image $x$. The goal is to match the maximum value in the response mapping $f(z, x)$ to the target location.

SiamRPN, SiamRPN++, and SiamBAN [19] all construct tracking frameworks based on Siamese, with the backbone network described in Section 2.1 as the $\varphi$ in this framework. We investigate the construction of an effective model $\theta$ using the more advanced ConvNext network as the backbone network to enhance the robustness and accuracy of tracking based on this framework.

### 2.3. Detection Model

Visual tracking tasks have many unique characteristics, but they still have much in common with target detection. Most advanced tracking methods follow the idea of target detection. For example, the RPN structure derives from Faster-RCNN. The RPN structure combined with the Siamese network achieves a surprising accuracy in SiamRPN and can solve the multi-scale tracking problem. This detection method using RPN structure and anchor design is known as anchor-based detector. According to the IOU threshold, the proposed boxes classify into positive and negative patches and obtain the exact target position by correcting the anchor using regression offsets [34,35]. However, these trackers require many anchors, resulting in unbalanced positive and negative samples and slow convergence during training. Moreover, the anchors introduce many hyperparameters, including the anchor's size, number, and aspect ratio [36,37], which leads to difficult training and requires heuristic adjustment. Therefore, anchor-free detection methods are developed, for instance, predicting the bounding box near the object's center [38] or detecting a set of opposite corners [39]. The most significant advantage of anchor-free is fast detection speed because it does not need to set anchors beforehand. It only regresses the target centroid and width and height of receptive field, dramatically reducing time consumption and arithmetic resources. On this inspiration, we elaborate an anchor-free visual tracking network.

## 3. Proposed Method

This section describes the details of CAFSN. Figure 3 illustrates its overall framework. A modified ConvNext network is used as the backbone network to extract image features. The feature fusion network applies a 3D C-Max fusion network to enhance the network's feature recognition capability by suppressing the responses of similar targets. A multi-branch prediction network comprises classification and regression branches for foreground and background classification and target size estimation.

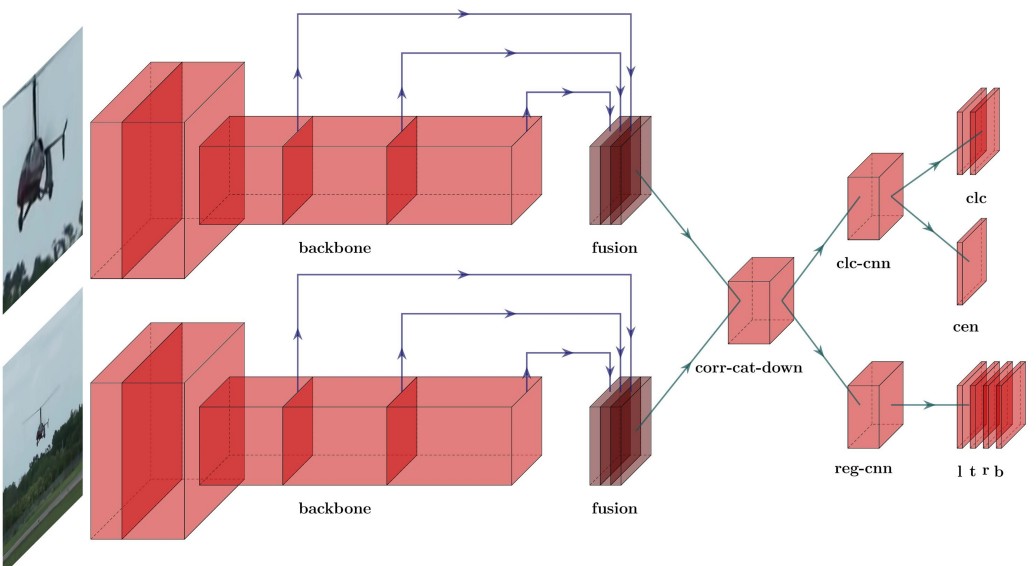

**Figure 3.** CAFSN model architecture. We define the post-fusion operation, the classification branch, and the regression branch as 'corr-cat-down', 'clc-cnn', and 'reg-cnn', respectively. 'clc' , 'cen' ,and 'ltrb' denote the output features of the classification branch, the central confidence branch and the regression branch, respectively.

### 3.1. Feature Extraction

The feature representation capability of the extraction network has an important impact on the accuracy and robustness of visual tracking. The ConvNext network has achieved state-of-the-art results in the field of image recognition and classification. However, if used directly in the tracking field, the effect of depth features extracted is not optimal. There is still a considerable gap in achieving progressive tracking performance. Focusing on this problem, we improve the ConvNext network inspired by the design guidelines proposed by SiamDW [40].

The network stride of Siamese tracker affects the accuracy of target localization. The receptive field size determines the ratio of the context information of the target in the feature to the local information of the target itself. Therefore, the network should be improved in terms of stride and receptive field, and Figure 4 shows the specific details of the improved ConvNext network. First of all, since the excessive strides will increase the error of target localization, the stem part of ConvNext is modified to increase the kernel size from 4 to 7 and reduce the stride size from 4 to 2 by adjusting the first convolution module. Secondly, To achieve the effect of the original stem to down-sample the image four times, max-pooling can reduce the feature map size while keeping the number of channels constant. Therefore, max-pooling is used for down-sampling. The paddings of sizes are three and max-pooling of stride sizes are 2. This step to pooling instead of convolution is beneficial to keep the network's translation invariability and fulfill the requirement of down-sampling stride. Each downsample in Figure 2 down-samples the image by a factor of 2, so the entire network down-samples up to 32 times, and the theoretical receptive field is 1688. The stride and receptive field sizes are too large for visual tracking. the downsample layer is need removed. Only add a convolution operation in Stage1 with a kernel size of 2 and stride of

2 to make the entire network down-sample by a factor of 8. At this point, the network's theoretical receptive fields are 807. In addition, the primary role of ConvBlock in the improved ConvNext network is to expand the number of channels, increase the number of network layers, and enhance the network's learning ability. Each ConvBlock uses deep separable convolution to reduce the calculated quantity of the network.

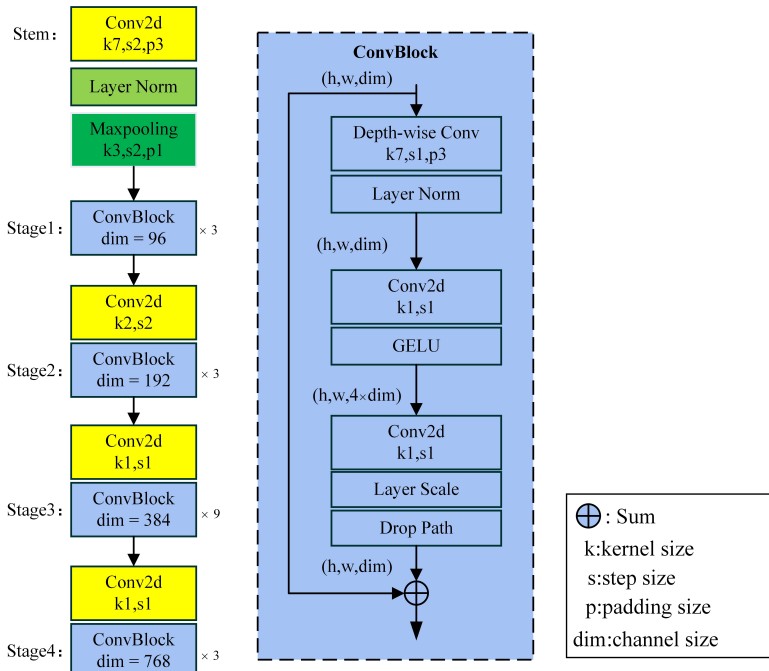

**Figure 4.** The specific details of improved ConvNext network. On the right is the specific detail of ConvBlock. The input and output sizes remain the same after passing a ConvBlock.

### 3.2. Feature Fusion and Enhancement

In deep learning, the essential information in shallow networks is beneficial for tracking localization, for instance, location, details, and edges. In contrast, features with more vital semantic information in deep networks are more critical for recognition. Thus, many methods merge low-level and high-level features to improve tracking accuracy [41]. We also consider joint multi-layer features to increase network representation. In addition, we find that the accuracy of target localization will be affected when similar targets are in the neighboring spatial region of the highest confidence position, resulting in jitter and offset between adjacent frames. Thus, we propose a novel 3D C-Max fusion module to solve this problem by strengthening the target features.

Convolution is a linear operation with translation invariance. It produces a high response to the target while producing high outputs for similar targets at different locations. This property has a favorable role in distinguishing foreground from background. However, there is a paradox of having a significant obstacle for discriminating similar targets, so in feature fusion, we use 3D C-Max fusion to augment the target's response and suppress the response of similar targets. Figure 5 shows the specific details of 3D C-Max fusion network. First, the input features are aligned consistently by crop and then via 3DMaxpooling to select the most responsive predictions in the local area to enhance the ability to distinguish similarity. The most important feature of this module is that it can be trained end-to-end by convolution and considers multidimensional features. As fusion in Figure 3, the feature maps generated by Stage2, Stage3, and Stage4 in the feature extraction network are used as the input data for feature fusion. The feature maps of these three layers have the same size, and only the number of channels is different. To eliminate the influence of the padding in the stem layer and ConvNext layer in the feature extraction network that destroys the translation invariance of the network, we crop the feature map size to $7 \times 7$ and obtain

the cropped feature map $x_c$. Then, we input $x_c$ to 3DMaxpooling to obtain output $y_f$. The output result $y_i$ corresponding to any point $x_i$ in the feature map $x_c$ is as follows:

$$y_i = \max_{k \in [s-p, s+p]} \max_{j \in N_i^{k \times k}} x_j \tag{2}$$

When max-pooling is performed once at a point $x_i$, $p$ denotes padding size and $k$ denotes the area size. As corr-cat-down in Figure 3, the output result $y_f$ is correlated and jointly spliced to obtain $\varphi(x, z)$ :

$$\varphi(x, z) = cat\left(F\left(y_{fi}(x), y_{fi}(z)\right)\right) \quad i \in 1, 2, 3 \tag{3}$$

where $F(\bullet)$ denotes the correlation operation, $F(y_{fi}(x), y_{fi}(z))$ denotes the correlation operation between the search image x and the output result of the template image z with a channel count of 256, and the number of channels of $\varphi(x, z)$ is $3 \times 256$. Finally, the number of channels $\varphi(x)$ is adjusted to 256 using $1 \times 1$ convolution to accomplish the entire feature fusion process, and Section 4.4 verifies the module's effect.

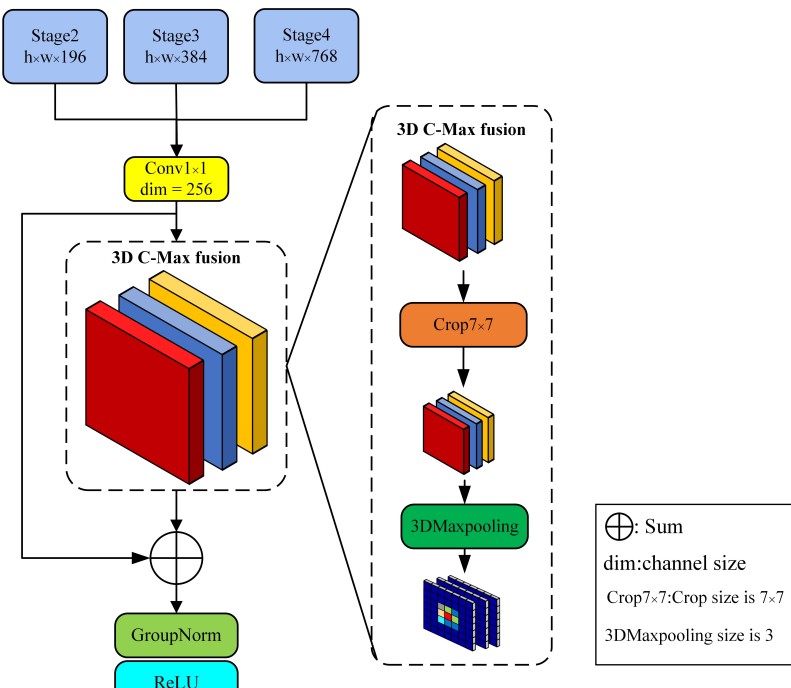

**Figure 5.** 3D C-Max fusion feature fusion network. When the input is a template image, $h = w = 15$. When the input is a search image, $h = w = 31$. Cropping only applies to template images.

### 3.3. Multi-Branch Prediction

The anchor-based trackers use RPN map through each point $(i, j)$ in the feature back to the original image $(x, y)$ and this point is used as the anchor center to generate multi-scale, multi-ratio anchor boxes as proposal boxes. Then, classify the proposal boxes into positive and negative samples according to the IOU threshold. Moreover, use these anchor boxes as references to regress the target bounding boxes. Distinctly, this paper classifies and regresses the receptive field area mapped back to the original image by each point in the feature graph. The tracker can carry out end-to-end training because no anchor is introduced, which avoids complex parameter adjustment and manual intervention during training and dramatically reduces the network's computation.

This paper decomposes the tracking task into two subtasks: The classification branch aims to predict the category at each location, and the regression branch aims to calculate the target bounding box at that location, i.e., the clc-cnn and reg-cnn branches in Figure 3.

Feature extraction network extracts features, and feature fusion network extracts features with dimension $R(h \times w \times c)$ $(h = 7, w = 7, c = 256)$. $h$ represents the length of the feature map. $w$ represents the width of the feature map. $c$ represents the channel number of the feature map. Through the classification branch, output a classification feature map with size $R_{clc}(h \times w \times 2)$. Any point $(i, j)$ in the feature map corresponds to a 2D vector representing the corresponding location's foreground and background scores in the input search region. The regression branch $\psi_{reg}$ outputs a regression feature map $R_{reg}(h \times w \times 4)$. Any point in the feature map corresponds to a 4D vector $(l, t, r, b)$ where $l$, $t$, $r$, and $b$ are the distances to the left, top, right, and bottom of the bounding box, respectively.

Specifically, the 4D vector $(l, t, r, b)$ corresponding to a point $(i, j)$ on the regression feature map output by the regression branch $\psi_{reg}$ is the distance from a point $(\lfloor \frac{s}{2} \rfloor + x_s, \lfloor \frac{s}{2} \rfloor + y_s)$ in the predicted input image to the four sides of the ground truth:

$$\begin{cases} x_l = \left(\lfloor \frac{s}{2} \rfloor + x_s\right) - l, \, y_t = \left(\lfloor \frac{s}{2} \rfloor + y_s\right) - t \\ x_r = \left(\lfloor \frac{s}{2} \rfloor + x_s\right) + r, \, y_b = \left(\lfloor \frac{s}{2} \rfloor + y_s\right) + b \end{cases} \tag{4}$$

where $(x_l, y_t)$ denotes the coordinates of the upper left point of the ground truth, and $(x_r, y_b)$ denotes the coordinates of the bottom right point of the ground truth. In classifications, a point $(i, j)$ of the classification feature map identifies as a positive sample when it maps back to the corresponding position $(\lfloor \frac{s}{2} \rfloor + x_s, \lfloor \frac{s}{2} \rfloor + y_s)$ on the input image, which is within the ground truth. Otherwise, it is considered a negative sample. $s$ is the complete step length of the feature extraction network, which is equal to 8, and the target value of the classification branch is calculated as follows.

$$Label_{clc} = \begin{cases} 1 & if \prod_{i=l,t,r,b} S(i) > 0 \\ 0 & otherwise \end{cases} \tag{5}$$

Expression $S(\cdot)$ determines whether the deviation between the distance $(l, t, r, b)$ from the point $(\lfloor \frac{s}{2} \rfloor + x_s, \lfloor \frac{s}{2} \rfloor + y_s)$ to the four edges of the ground truth, and the bounding box is within the distance range of $(1-\alpha)$-times as long and $(1-\alpha)$-times as wide. $\alpha$ is a constant, and its range can be set between [0,1], which is 0.6 in this paper.

$$S(l, t, r, b) = \begin{cases} 1 & if \begin{cases} l, r > \alpha \times \left(\frac{x_r - x_l}{2}\right) \\ t, b > \alpha \times \left(\frac{y_b - y_t}{2}\right) \end{cases} \\ 0 & otherwise \end{cases} \tag{6}$$

Therefore, this paper uses the cross-entropy as the loss function of the classification branch, which calculate as follows

$$L_{clc} = -\frac{1}{2} \left( \sum_{pos} R_{clc} + \sum_{neg} R_{clc} \right) \tag{7}$$

The formula *pos* represents the position where the result is 1 in the $Label_{clc}$ and *neg* represents the position where the result is 0 in the $Label_{clc}$. Then, calculate the IOU [42] between the ground-truth bounding box and the predicted bounding box, and finally obtain the loss function of the regression branch.

$$L_{reg} = \frac{1}{\sum Label_{clc}(i,j)} \sum_{i \in h, j \in w} \left( Label_{clc}(i,j) \right. \\ \left. \times L_{IOU}\left( R_{reg}(i,j,:), [x_l, y_t, x_r, y_b] \right) \right) \tag{8}$$

Jiang et al. [43] showed that classification confidence and localization accuracy do not correlate well. Suppose we do not consider the quality of target state estimation. In that case, the classification score is directly used to select the final regression box, leading to a decrease in localization accuracy. According to the analysis of Luo [44] et al., the pixels near the center

of the ground truth are more critical than other pixels. Therefore, the positions farther from the center are prone to producing a low-quality prediction bounding box, which reduces the tracking system's performance. In order to suppress low-quality prediction boxes and remove outliers, we add a simple and effective quality assessment branch and redefine the center's confidence. As shown in Figure 3, the center's confidence branch outputs a center confidence feature map $R_{cen}(h \times w \times 1)$. For a point $(i, j)$ on the feature map output by the central confidence branch, this paper defines it as follows:

$$C_{cen} = Label_{clc} \times \left( 1 - \sqrt{\frac{(l-r)^2 + (t-b)^2}{(l+r)^2 + (t+b)^2}} \right) \tag{9}$$

where $C_{cen}$ is the normalized distance from the corresponding position $(x, y)$ in the search area to the center of the target, and $C_{cen}$ is equal to 0, when $(x, y)$ is background, so the loss function of the center confidence is as follows:

$$
\begin{aligned}
L_{cen} = {} & \frac{1}{\sum Label_{clc}(i,j)} \sum_{Label_{clc}(i,j)=1} \{ (1 - C_{cen}(i,j)) \\
& \times \log(1 - R_{cen}(i,j)) + C_{cen}(i,j) \times \log R_{cen}(i,j)
\end{aligned} \tag{10}
$$

Finally, the total loss function of the network is $L = L_{clc} + \lambda_1 L_{cen} + \lambda_2 L_{reg}$. The formula $L_{clc}$ is the classification loss function, $L_{cen}$ is the central confidence loss function, $L_{reg}$ is the regression loss function, $\lambda_1$, and $\lambda_2$ are the weight of the central confidence loss function and regression loss function, respectively.

## 4. Experiments

The CASFN is implemented based on Pytorch on an Intel(R) Xeon(R) Gold 6248R CPU @ 3.0GHz, 192GB of RAM (Intel, Santa Clara, CA, USA), and a Tesla V100 NVIDIA GPU with 32GB of RAM (NVIDIA, Santa Clara, CA, USA). We adopt ILSVRC-VID/DET [45], COCO [46], and GOT-10k [47] as our base training datasheets and select frame pairs as input data in intervals less than 100. Then we perform data augmentation by uniformly distributed random movement and scaling the search image, setting the input search image and template image size to 255 and 127 pixels, respectively. The modified ConvNext-tiny as the backbone network, pretraining the network on ImageNet in ImageNet [45], and then retrain our model using the parameters as initialization.

During the training process, The training parameters are shown in the following Table 1. The batch size is set as 128, and 20 epochs are conducted using The random gradient descent method with a warm-up strategy for training. The initial learning rate is 0.001, which reaches the maximum of 0.005 in the fifth epoch and decreases to 0.0005 in logarithmic form in the following 15 epochs. In the first ten epochs, the backbone network is frozen, and the classification and regression branches are trained. Then unfreeze and train the entire network. The total training time takes around 40.82 h.

**Table 1.** The main training parameters.

| Parameter | Value |
| --- | --- |
| Template image size | 127 |
| Search image size | 255 |
| Learn rate | 0.001 |
| Batch size | 128 |
| Epoch number | 20 |
| Start learn rate | 0.005 |
| End learn rate | 0.0005 |
| Weight of the central confidence | 3 |
| Weight of the regression branches | 1 |
| Output feature map size | 25 |

Figure 6 shows the specific procedure details of the test. The initial frame is input as the template image in the testing process, and the next frame is taken as the search image. The classification score *clc*, center confidence *cen*, and prediction width and height $(l + r, t + b)$ are output through the CAFSN. The cosine window *w* is used to punish the score of the edge region far from the center point. The scale change penalty *s* is introduced to reorder the classification scores, select the coordinates of the maximum value, and calculate the relative displacement, where we have the following.

$$(x_o, y_o) = \arg\max\{(1 - \lambda)clc \times cen \times s + \lambda w\} \tag{11}$$

Then, the relative displacement is added to the coordinates of the target's center point in the previous frame to obtain the center point of the current target. The corresponding width and height prediction values are selected from the regression branch according to the predicted maximum coordinate. Finally, this paper tests the comprehensive performance of the tracker on the OTB, UVA, and GOT10k test datasheets. It carries out ablation experiments to analyze the functions and effects of each module.

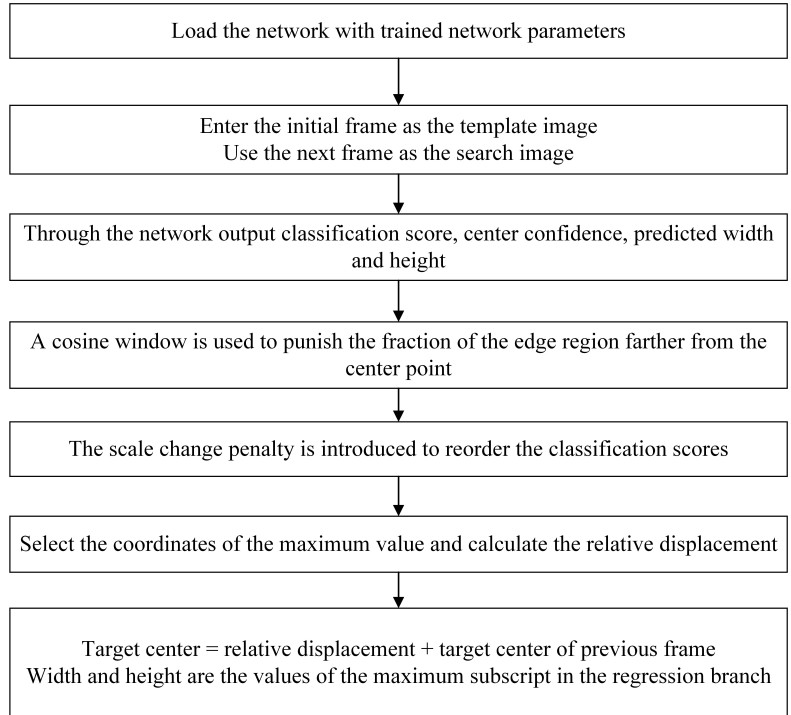

**Figure 6.** Testing Details. The testing phase uses the offline tracking strategy.

*4.1. Experiments on OTB100*

The complete OTB100 benchmark contains 100 sequences that address the 11 attributes of target tracking, including illumination variation (IV), scale variation (SV), occlusion (OCC), deformation (DEF), motion blur (MB), fast motion (FM), in-plane rotation (IPR), out-of-plane rotation (OPR), out-of-view (OV), background clutter (BC), and low resolution (LR). The sequences include grayscale and color images, and each video sequence contains at least two attributes. In this paper, each tracker's accuracy and success rate in one-pass evaluation (OPE) are evaluated on the OTB dataset against ten other progressive methods, including SiamRPN and Ocean [48]. The accuracy evaluation is the probability that the estimated center position is within 20 pixels of the actual center position. The success rate defines the total number of successful frames as a percentage of all frames. The overlap rate defines the intersection of the tracker's estimation frame and the ground-truth bounding box. The frame is considered to track success when it is more significant than a set overlap rate threshold. The overlap rate takes in the range of 0 to 1.

Then, the success rate graph is plotted for the threshold change from 0 to 1. The success rate graph is evaluated with the index area under the curve (AUC). Figure 7 shows the comprehensive performance evaluation results on the OTB100 dataset, and the CAFSN ranks third in both metrics combined. In particular, as shown in Figure 8, CAFSN shows more outstanding performance in terms of low resolution and background clutter. The CAFSN ranks first in the low-resolution cases, of which AUC reaches 69.1% and precision attains 98.2%. The CAFSN ranks second in the background clutter cases, of which AUC reaches 66.7% and precision attains 92.4%. The result indicates that CAFSN has an excellent performance in anti-noise. It is attributed to the feature fusion strategy that enhances the semantic information of the target.

Figure 9 shows a qualitative comparison of our method with Ocean, SiamDWrpn, and DeepSRDCF on challenging sequences. The five sequences contains same challenging attributes, i.e., BC and LR. These trackers mostly perform well on these sequences. However, our tracker has better performance on background clutters and low resolution. Benefiting from suppressing the background, our tracker can perform well on long sequences ('Dudek').

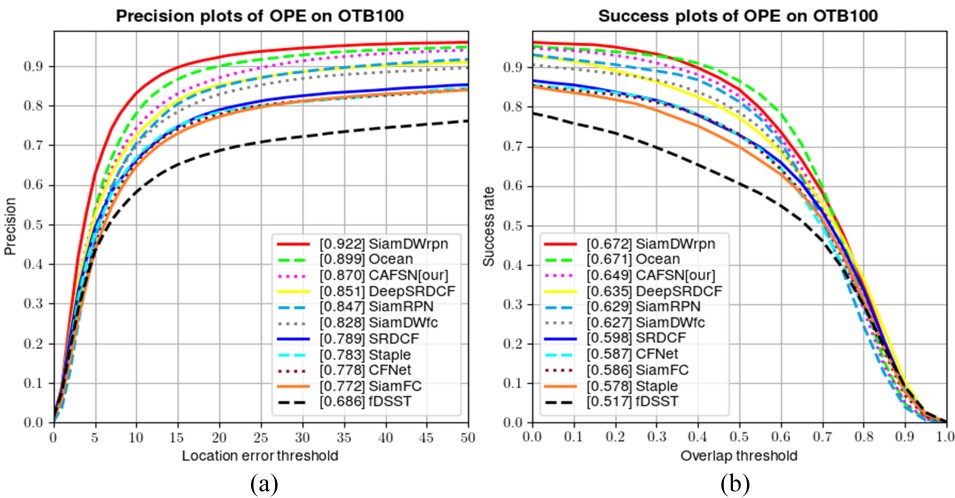

**Figure 7.** Comprehensive network performance evaluation results based on OTB100 dataset. (**a**) is the precision. (**b**) is the success.

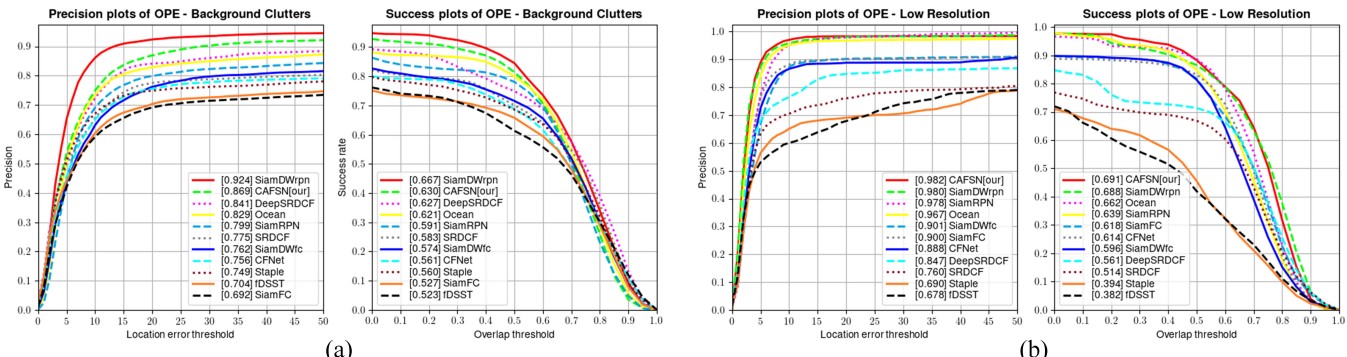

**Figure 8.** Results based on the OTB100 dataset in the background clutter, low-resolution case. (**a**) is the precision and success in background clutter case. (**b**) is the precision and success in low-resolution case.

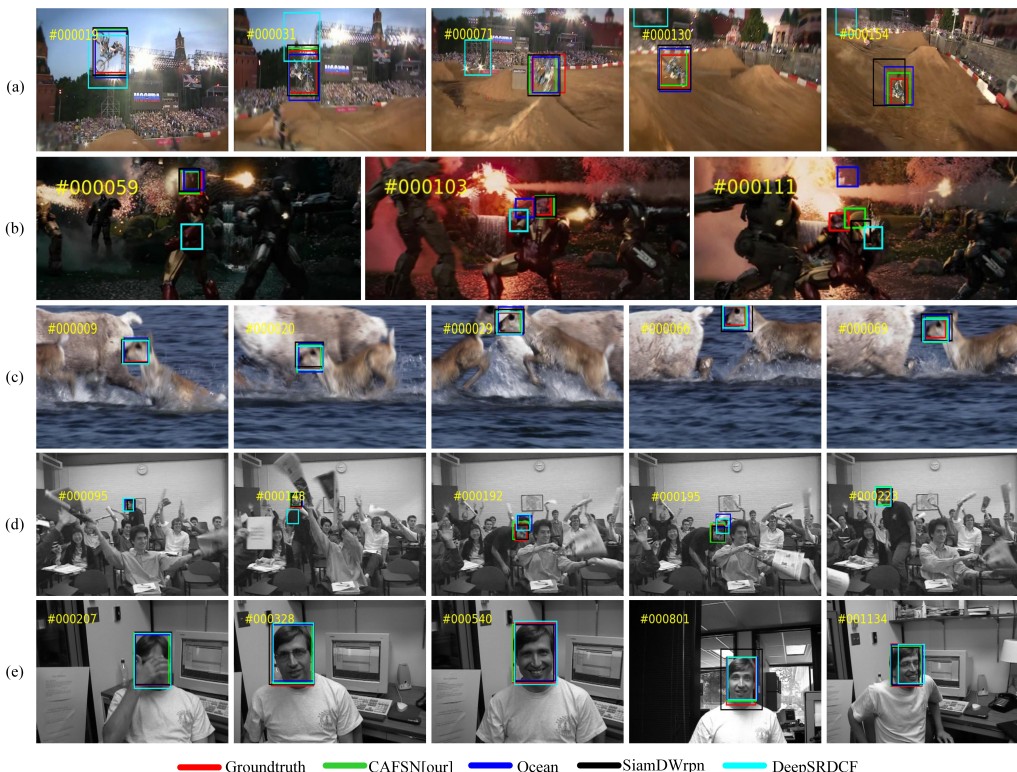

**Figure 9.** Qualitative comparisons with several trackers on some challenging sequences in OTB: (**a**) MotorRolling, (**b**) Ironman, (**c**) Deer, (**d**) Freeman4, and (**e**) Dudek.

### 4.2. Experiments on UAV123

The UAV123 dataset constructs a high-definition video sequence obtained from the shooting angle of the UAV, containing 123 video sequences with more than 110K frames in total. The evaluation metrics are consistent with OTB100, but the scale of the rectangular box of the target in the video sequence of this dataset will vary more, so it will be more demanding for the tracker to adapt to the scale of the target. UAV123 mainly involves 13 tracking attributes, including aspect ratio variation, background clutter, camera motion, fast motion, complete occlusion, illumination variation, low resolution, out-of-view, partial occlusion, similar targets, and scale variation. This paper compares the UAV123 dataset with eight other advanced methods, including SiamRPN++, SiamBAN, etc. Figure 10 shows the comprehensive performance evaluation results on the UAV123 dataset. The CAFSN is ranked third overall in this accuracy metric and second in the success rate metric. In particular, as shown in Figure 11, CAFSN significantly raises the tracking accuracy in terms of aspect ratio variation and similar targets. The CAFSN ranks first in the aspect ratio variation and similar target cases, of which the AUC reaches 69.6% and 68.5%, and precision attains 76.8%, 76.3%. The result manifests that the proposed 3D C-Max fusion module enhances the network's ability to discriminate similar objects while excellently discriminating the foreground from background. The CAFSN ranks second in the full occlusion and partial occlusion cases, of which AUC reaches 44.7% and 63.8%, and precision attains 61.4% and 73.3%. The experiment result shows that the proposed 3D C-Max fusion module can also strengthen the target features and increase the tracking robustness of the target.

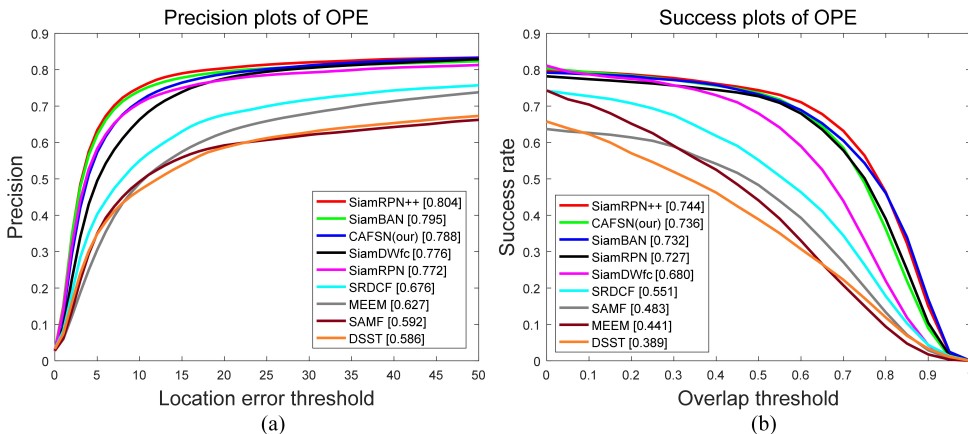

**Figure 10.** Comprehensive network performance evaluation results based on UAV123 dataset. (**a**) is the precision. (**b**) is the success.

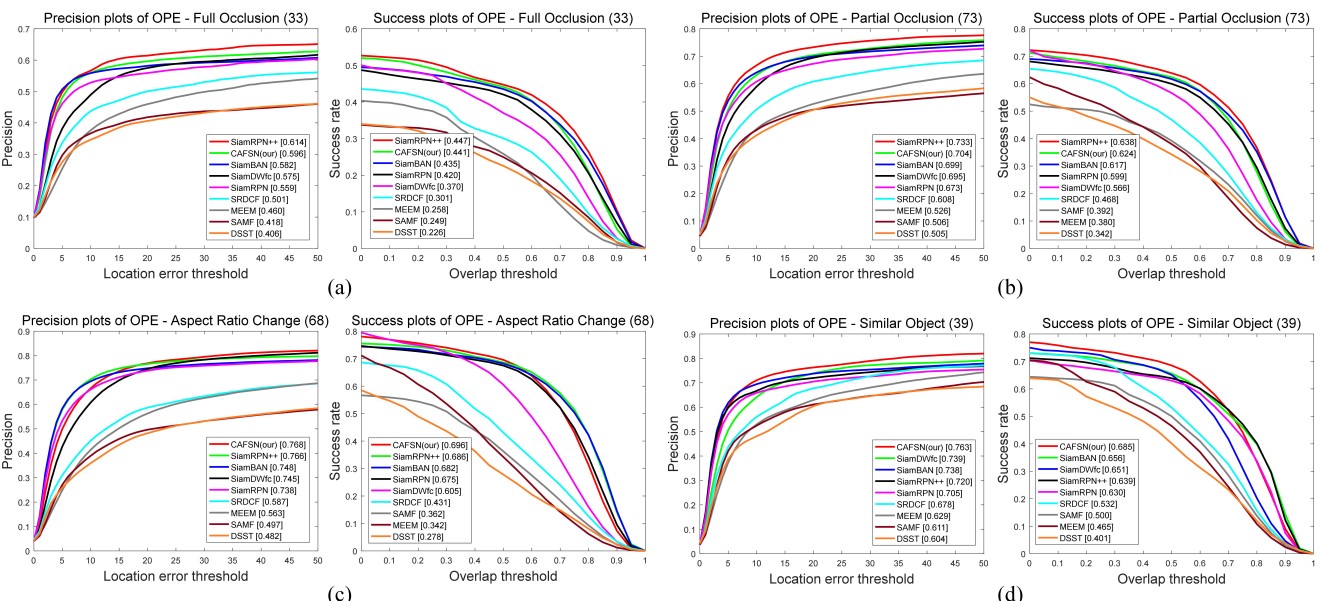

**Figure 11.** Comprehensive network performance evaluation results based on the UAV123 dataset with (**a**) full occlusion, (**b**) partial occlusion, (**c**) varying aspect ratios, and (**d**) similar targets.

### 4.3. Experiments on GOT10K

GOT-10k contains over 10,000 videos with over 1.5 million manually labeled bounding boxes. It comprises 563 target categories and 87 motion patterns. The dataset has zero overlaps with the training set, and the provided evaluation metrics include success rate graph, average overlap (AO), frames per second (FPS), and degree success rate (SR). The AO indicates the average overlap between estimated bounding and ground truth boxes. $SR_{0.5}$ represents the ratio of successful frame tracking with more than 0.5 overlaps, and $SR_{0.75}$ represents successful tracking with more than 0.75 overlaps. FPS represents the maximum number of frames per second that the algorithm can process in this dataset. It can be used to represent the computational complexity of an algorithm and reflect the real-time performance of the algorithm. We evaluate the proposed algorithm on GOT-10k and compared it with 13 progressive methods such as SiamRPN++, ATOM [49]. The test set embodies 84 object classes and 32 motion classes with 180 video segments, allowing efficient evaluations. Figure 12 shows the success rate graph of each algorithm on GOT-10K, and the performance of the proposed algorithm in this paper is ranked third. Table 2 shows the comparison details of different metrics. Although CAFSN is slightly weaker than ATOM and SiamRPN++ in terms of performance, our algorithm has a massive advantage

in real-time tracking, approximately 2 × faster than ATOM. This result verifies that CAFSN has less computation and is simple.

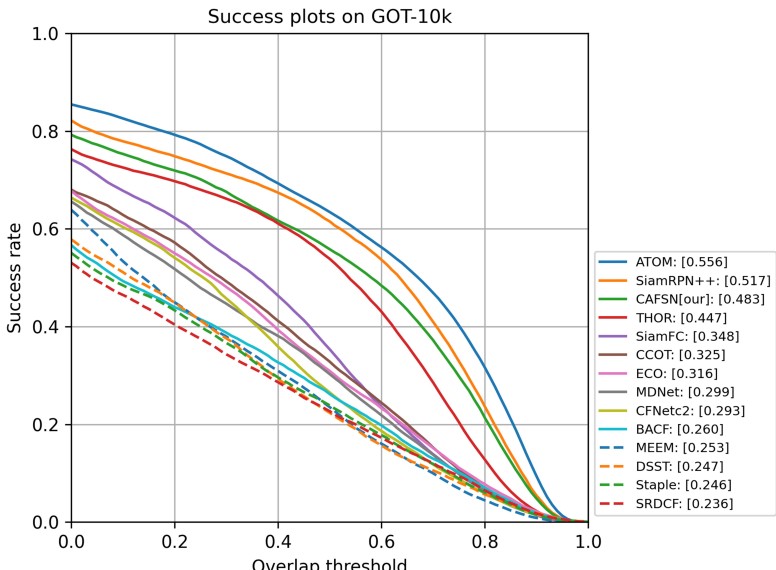

**Figure 12.** Comprehensive network performance evaluation results based on GOT-10K dataset.

**Table 2.** Specific data on the evaluation metrics of each tracker on the GOT-10K dataset. Red, Blue and Green fonts indicate the top-3 trackers, respectively. The bolded part is our approach.

| Tracker | AO | $SR_{0.5}$ | $SR_{0.75}$ | FPS |
|---|---|---|---|---|
| SRDCF | 0.236 | 0.227 | 0.094 | 5.58 |
| Staple | 0.246 | 0.239 | 0.089 | 28.87 |
| DSST | 0.247 | 0.223 | 0.081 | 18.25 |
| MEEM | 0.253 | 0.235 | 0.068 | 20.59 |
| BACF | 0.26 | 0.262 | 0.101 | 14.44 |
| CFNetc2 | 0.293 | 0.265 | 0.087 | 35.62 |
| MDNet | 0.299 | 0.303 | 0.099 | 1.52 |
| ECO | 0.316 | 0.309 | 0.111 | 2.62 |
| CCOT | 0.325 | 0.328 | 0.107 | 0.68 |
| SiamFC | 0.348 | 0.353 | 0.098 | 44.15 |
| THOR | 0.447 | 0.538 | 0.204 | 1.00 |
| **CAFSN (our)** | 0.483 | 0.558 | 0.298 | 58.44 |
| SiamRPN++ | 0.517 | 0.615 | 0.329 | 3.18 |
| ATOM | 0.556 | 0.634 | 0.402 | 20.71 |

## *4.4. Ablation Experiment*

To verify the effectiveness of each component of our tracker, we implement several ablation experiments evaluated on the OTB dataset, and Table 3 shows the detailed evaluation results. We start the tuning training from the dataset, firstly using ConvNext as the backbone network and acquiring 64K image pairs on the GOT-10k dataset for training, followed by training on four datasets of ILSVRC-VID/DET, COCO, LaSOT, and GOT-10k, with an AUC improvement of 2.6% and precision improvement of 3.8%. In order to improve the target response value and suppress the effect of similar targets, the 3D C-Max Fusion proposed in this paper is used, resulting in a 1.7% improvement in AUC and a 0.5% improvement in precision. In Figure 13, the y-label indicates the sequence name of the OTB dataset, and the x-label indicates the output results of each component. The first three columns visualize the features extracted by the backbone network, the features output by the feature fusion network and the results output by the multi-branch prediction network, respectively. The last column shows the effect after superimposing the visualized predic-

tion results onto the original image, highlighting the target more, improving the target response value, and enhancing the discrimination ability with similar targets under the effect of 3D C-Max Fusion. To suppress low-quality prediction frames and remove outliers, we designed a simple but effective quality assessment branch, which resulted in a 1.2% improvement in AUC and a 1.5% improvement in precision. Finally, 600 k image pairs were collected on four datasets for training and parameter optimization, achieving an excellent performance of 64.9% AUC and 87% precision in the OTB dataset.

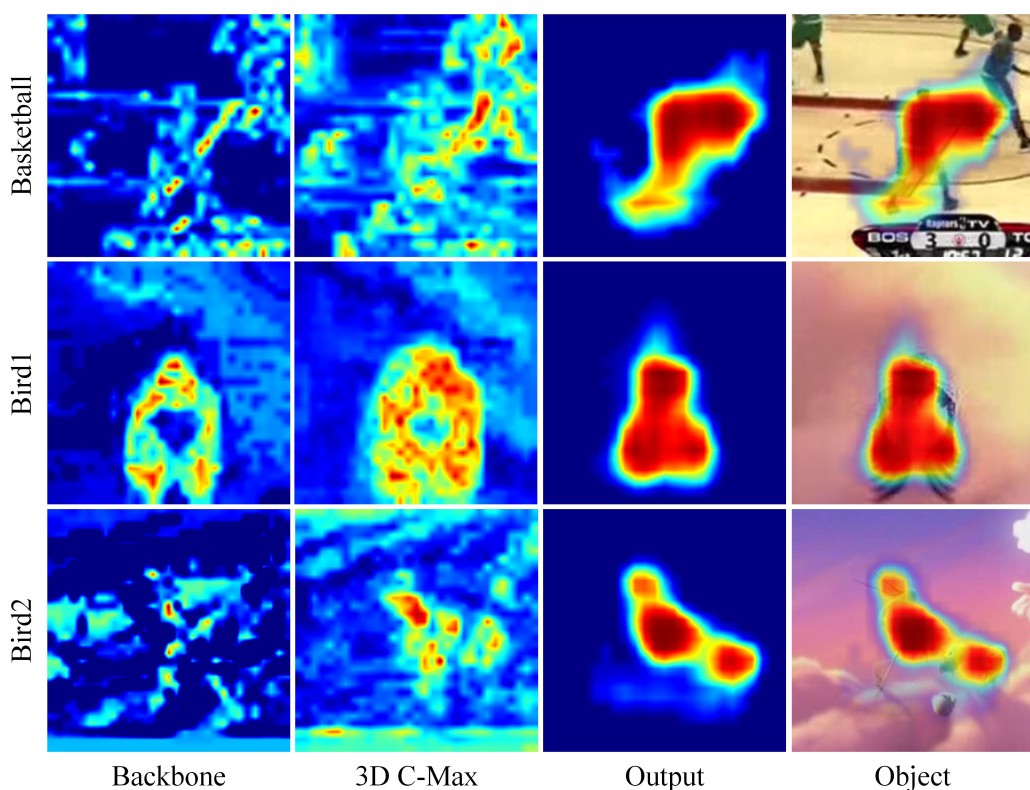

**Figure 13.** Visual response maps. The higher the response, the more salient the results.

**Table 3.** Ablation experiments performed on the OTB dataset. '4 Data' Indicates that the number of training datasets is 4. 'Improved' means using an improved ConvNext network. '64 K' and '600 K' indicate the number of image pairs. The bolded part is the method finally adopted.

| Datasheets | Backbone | 3D C-Max | Center Loss | Image Pairs | AUC | Precision |
|---|---|---|---|---|---|---|
| GOT-10k | ConvNext | NO | NO | 64 K | 0.394 | 0.566 |
| 4 Data | ConvNext | NO | NO | 64 K | 0.42 | 0.604 |
| 4 Data | Improved | NO | NO | 64 K | 0.52 | 0.736 |
| 4 Data | Improved | Yes | NO | 64 K | 0.537 | 0.741 |
| 4 Data | Improved | Yes | Yes | 64 K | 0.549 | 0.756 |
| **4 Data** | **Improved** | **Yes** | **Yes** | **600 K** | **0.649** | **0.870** |

To demonstrate the performance advantages of the proposed approach, we performed a contrast ablation experiment on OTB100. The training dataset for these networks is 64 K image pairs obtained from the four datasets mentioned above. The experimental results are shown in Table 4. We compare the backbone network and the proposed 3D-CMax module with other similar approaches. ResNet50 was selected as the backbone network and Context Enhancement Module (CEM) as the enhancement module. Without augmentation, Backbone improves 0.7% (0.525 vs. 0.532) in AUC and 0.9% (0.716 vs. 0.725) in accuracy using the improved ConvNext over using ResNet50. When Backbone uses the improved ConvNext, the enhancement module uses 3DCMax to improve 0.7% (0.542 vs. 0.549)

in AUC and 0.4% (0.751 vs. 0.756) in accuracy over using CEM. When Backbone uses ResNet50, and 3DCMax improves 0.5% (0.538 vs. 0.543) in AUC and 0.9% (0.743 vs. 0.752) in accuracy over using CEM.

**Table 4.** Ablation study of the backbone and fusion network on OTB100. ResNet-50: Residual Network with 50 layers; CEM: Context Enhancement Module; AUC: Area Under the Curve.

| Network | AUC | Precision |
|---|---|---|
| ResNet50 | 0.525 | 0.716 |
| Improved ConvNext | 0.532 | 0.725 |
| ResNet50 + CEM | 0.538 | 0.743 |
| Improved ConvNext + CEM | 0.542 | 0.751 |
| ResNet50 + 3D-CMax | 0.543 | 0.752 |
| Improved ConvNext + 3D-CMax (Our) | 0.549 | 0.756 |

## 5. Conclusions

We propose a new anchor-free high-performance visual tracking network architecture, CAFSN, with a tidy, complete convolutional network. Our CAFSN overcomes the model's drifts and tracking failure in complex tracking scenes, such as low-resolution, background clutter, aspect ratio variation, similar target, full occlusion, partial occlusion, and more. The advanced ConvNext network is improved to obtain a backbone network with superior characterization capability. We combine multi-layer features and propose 3D C-Max Fusion to solve similar target interference problems. Then, we designed a central confidence branch based on Euclidean distances to remove outliers to suppress low-quality prediction frames in the prediction network. CAFSN has a simple and effective structure and achieves advanced tracking results on the OTB100, UVA123, and GOT-10K datasets, proving that CAFSN has high noise immunity and a high ability to distinguish between similar targets.

The current network architecture facilitates the continuation of optimizing the depth and structure of the network and adjusting module architecture in pursuit of higher tracking performance and a more concise network structure. In order to reduce computational complexity, this method does not update the template comparing to ATOM. This causes tracking to fail easily in the next frame when the target is lost in this frame during long tracking sessions. In the future, a suitable and convenient template update pattern needs to be studied for ensuring the balance of tracking accuracy and complexity.

**Author Contributions:** Conceptualization, Q.X. and Z.Z.; methodology, Q.X., H.D. and Z.Z.; software, Q.X. and Y.L.; validation, Q.X. and G.L.; formal analysis, Q.X. and H.D.; investigation, Q.X. and Y.L.; resources, Q.X., H.D. and Y.L.; data curation, Q.X.; writing—original draft preparation, Q.X. and X.R. All authors have read and agreed to the published version of the manuscript.

**Funding:** This research received no external funding.

**Acknowledgments:** We are grateful to the High Performance Computing Center of Central South University for assistance with computations.

**Conflicts of Interest:** The authors declare no conflict of interest.

## Abbreviations

The following abbreviations are used in this manuscript:

| | |
|---|---|
| CFNet | CFNet: Cascade and Fused Cost Volume for Robust Stereo Matching; |
| SiameseFC | Fully Convolutional Siamese Networks; |
| SiamRPN | High Performance Visual Tracking with Siamese Region Proposal Network; |
| DaSiam | Distractor-aware Siamese Networks for Visual Object Tracking; |
| CSiam | Siamese cascaded region proposal networks for real-time visual tracking; |
| SiamRPN++ | SiamRPN++: Evolution of Siamese Visual Tracking with Very Deep Networks; |
| SiamBAN | Siamese box adaptive network for visual tracking; |

|           |                                                            |
|-----------|------------------------------------------------------------|
| SiamDW    | Deeper and Wider Siamese Networks for Real-Time Visual Tracking; |
| Ocean     | Ocean: Object-aware Anchor-free Tracking;                  |
| ATOM      | ATOM: Accurate Tracking by Overlap Maximization;           |
| DeepSRDCF | Convolutional Features for Correlation Filter Based Visual Tracking; |
| TransT    | Transformer tracking;                                      |
| SwinTrack | SwinTrack: A Simple and Strong Baseline for Transformer Tracking. |

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
