# Peer review of "A ConvNext-Based and Feature Enhancement Anchor-Free Siamese Network for Visual Tracking"

_electronics, doi:10.3390/electronics11152381_

Round 1

Reviewer 1 Report

1. The novelty and new outcomes of the present work are not sufficient for this high impact factor journal. There are many similar works and could not find any significant difference.

2. There is no discussion of user requirements, technological options and support for the decisions made at the design. The authors should include more technical details and explanations.

3. More experiments and some comparisons with other up-to-date methods should be addressed or added to back your claims to expand your experiments and analysis of results further.

4. The authors need to interpret the meanings of the variables. Some parameters and their values are unknown. It would be better to show all these parameters and explain the reason for those numbers in the table.

5. The experiment results show the performance with high accuracy, please show the parameter settings of each approach using a table.

6. How about the computation complexity of the proposed method compared with related work?

7. The comparison to other improved schemes (more current literature in the area) is required, such as [A]. This paper should summarize those results and give a comprehensive performance comparison with previous works.

8. The conclusion and future work part can be extended to have a better understanding of the approach and issues related to that which can be taken into consideration for future work.

[A] M. Liu, J. Ma, Q. Zheng, Y. Liu and G. Shi, 3D Object Detection Based on Attention and Multi-Scale Feature Fusion, Sensors, vol. 22, no. 10, 2022.

Author Response

Please see the attachment. The manuscript shows the revised paper. If you want to see the tracks of change in the manuscript. Please annotate the command, "\usepackage[final]{changes}", and then add the command, \usepackage{changes}.

Reviewer 2 Report

This manuscript proposes a fusion network for visual tracking and determining the response targets of similar objects using multi-layer features. The proposed method was evaluated using different benchmarked databases, which obtained good results. However, it lacks in-depth explanations of the equations. Therefore, the authors must explain the variables in more detail in all the equations. Besides, the technical writing of the paper needs to be enhanced. The contributions of this paper (L79-L88) should be analyzed and validated obtained results compared to other researchers

In addition to the following comments:

-Enhance the abstract and the conclusion to reflect the contribution of the work.

- Add more literature for review, and the results need to be contrasted and compared with the results of the other state-of-the-art methods in the literature to support the main findings thoroughly.

- Fig.4 unclear what information is presented. It is hard for the reader to understand it. So, please modify it to be more readable and informative.

- Fig.5 unclear what is the meaning of the second part. Also, the first arrow is divided into connector steps according to what condition. So, please modify it to be more readable and informative.

-The research paper should be written in the third person's perspective; words such as "we", "our," etc., must be avoided.

-Avoid using many references together, such as L32 [6-8], L43 [15-17], L97, etc. You should classify the studies and write a paragraph about each study or category.

- Do not use short terms without a prior definition (AUC, 3D C-Max, SiamRPN, RPNs, DaSiam, CSiam, SwinTrack), etc.

 -Too-long sentences make the meaning unclear. Consider breaking it into multiple sentences—for example, L24-26, L33-35, L51-54, L54-56, L71-74, etc.

- It is unclear what methodology was used and how they implemented it. Details on the methodological approach adopted should be defined precisely.

- It needs more comprehensive evaluations and comparisons with other researchers (mentioned in the related work) to validate the obtained results supported by graphical and tabular data.

- Some grammatical or spelling errors that make the meaning unclear, sentence construction errors, and punctuation errors. The following are some examples:

L21: still a challenging problem in practical  …….should be …..    still challenging in practical

L34: utilizes Siamese network ….should be …..    utilizes the Siamese network

L96: using Siamese network   ….should be …..     using the Siamese network

 L99: where Vision Transformer (ViT) [28,29], as the backbone network, has achieved excellent performance. Unclear sentence and need to be rephrased.

 L109: The role of stem and downsample   ….should be ….. The role of the stem and downsample

Author Response

(The authors gave the same response as above.)

Reviewer 3 Report

In this paper, a new anchor-free high-performance visual tracking network architecture, CAFSN, with a tidy, complete convolutional network is proposed. From my view, this paper is well organized and the proposed method is valuable for this research filed. After reviewed this paper, there are some questions and suggestions as follows.

  1. The literature review must be enhanced. You must review all significant similar works that have been done. Also, review some of the good recent works that have been done in this area and are more similar to your paper.
  2. It is necessary to experimentally analyze the proposed algorithm in terms of time consumed and compare with other algorithms.
  3. What are the advantages and disadvantages of this study compared to the existing studies in this area? This needs to be addressed explicitly and in a separate subsection.
  4. There are many grammatical mistakes and typo errors. For example, "Figure 12. Visualiza response maps...."

Author Response

Please see the attachment. The manuscript shows the revised paper. If you want to see the tracks of change in the manuscript, please annotate the command, "\usepackage[final]{changes}", and then add the command, \usepackage{changes}.

Round 2

Reviewer 1 Report

This paper has edited and revised according to the reviewer's suggestions.

Reviewer 2 Report

The revised manuscript is enhanced to the level that can be published upon the editor's opinion.

Reviewer 3 Report

Good revisions have been made in the paper and the revised version has the necessary qualities for acceptance compared to the previous version. In my opinion, the article is acceptable in its current form.